# β3 Adrenergic Receptor Agonist Mirabegron Increases AQP2 and NKCC2 Urinary Excretion in OAB Patients: A Pleiotropic Effect of Interest for Patients with X-Linked Nephrogenic Diabetes Insipidus

**DOI:** 10.3390/ijms24021136

**Published:** 2023-01-06

**Authors:** Serena Milano, Fatima Maqoud, Monica Rutigliano, Ilenia Saponara, Monica Carmosino, Andrea Gerbino, Giuseppe Lucarelli, Michele Battaglia, Maria Svelto, Giuseppe Procino

**Affiliations:** 1Department of Biosciences, Biotechnology and Biopharmaceutics, University of Bari, 70125 Bari, Italy; 2Urology, Andrology and Kidney Transplantation Unit, Department of Emergency and Organ Transplantation, University of Bari, 70124 Bari, Italy; 3Department of Science, University of Potenza, 85100 Potenza, Italy

**Keywords:** AQP2, NKCC2, mirabegron, beta3-adrenoreceptor, nephrogenic diabetes insipidus, diuresis, vasopressin

## Abstract

We previously reported the novel finding that β3-AR is functionally expressed in the renal tubule and shares its cellular localization with the vasopressin receptor AVPR2, whose physiological stimulation triggers antidiuresis by increasing the plasma membrane expression of the water channel AQP2 and the NKCC2 symporter in renal cells. We also showed that pharmacologic stimulation of β3-AR is capable of triggering antidiuresis and correcting polyuria, in the knockout mice for the AVPR2 receptor, the animal model of human X-linked nephrogenic diabetes insipidus (XNDI), a rare genetic disease still missing a cure. Here, to demonstrate that the same response can be evoked in humans, we evaluated the effect of treatment with the β3-AR agonist mirabegron on AQP2 and NKCC2 trafficking, by evaluating their urinary excretion in a cohort of patients with overactive bladder syndrome, for the treatment of which the drug is already approved. Compared to baseline, treatment with mirabegron significantly increased AQP2 and NKCC2 excretion for the 12 weeks of treatment. This data is a step forward in corroborating the hypothesis that in patients with XNDI, treatment with mirabegron could bypass the inactivation of AVPR2, trigger antidiuresis and correct the dramatic polyuria which is the main hallmark of this disease.

## 1. Introduction

We previously reported that the type-3 β-adrenoreceptor (β3-AR) is expressed in epithelial cells lining specific segments of the kidney tubule, including the cortical and outer medullary collecting duct (CD) and the thick ascending limb of the Henle’s loop (TAL) [1,2]. Interestingly, we also showed that, in these kidney tubules, β3-AR shares both the basolateral localization and the cAMP-eliciting intracellular transduction pathway with the type-II vasopressin (AVP) receptor (AVPR2) [3], which physiological stimulation triggers antidiuresis in mammals [4,5]. Incubation of mouse vital kidney slices with the β3-AR selective agonist BRL37344 [6], resulted in accumulation of the water channel aquaporin-2 (AQP2) at the apical membrane of CD epithelial cells and phosphorylation/activation of the NKCC2 symporter in an AVP-like fashion [1]. Of note, these responses are physiologically elicited in vivo by AVP and result in increased NaCl tubular reabsorption, the consequent generation of the cortico-medullary osmotic gradient, and eventually water reabsorption in the CD, allowing urine concentration, namely antidiuresis [7]. Furthermore, in vivo, intraperitoneal administration of BRL37344 triggered antidiuresis in AVPR2 conditional knockout mice [8], which are unable to concentrate urine in response to AVP, recapitulating the main hallmark of the rare human genetic disease X-linked nephrogenic diabetes insipidus (XNDI) [1]. This evidence has interesting physiological implications, as it reveals a possible role of the sympathetic nervous system in sustaining the classical AVP-mediated antidiuresis and, therefore, represents a potential tool to restore antidiuresis in XNDI patients with inactivating mutations in the AVPR2 gene responsible for polyuria, low urine osmolality and polydipsia [9].

To push in the direction of a possible clinical trial on the efficacy of β3-adrenergic stimulation to correct the polyuria of XNDI patients, we also demonstrated that the human β3-AR is resistant to agonist-induced desensitization in renal cells in vitro, the first step to hypothesize a long-lasting efficacy on the kidney concentrating ability of a chronic treatment with a β3-AR agonists [10].

Here, to obtain the proof-of-principle that also in the human kidney the pharmacological agonism of the β3-AR promotes the trafficking of AQP2 and NKCC2, we analyzed the 24 h urine of a cohort of patients with overactive bladder syndrome (OAB) [11,12,13], with apparent normal renal function, before and during chronic treatment (up to 12 weeks) with the human β3-AR agonist mirabegron [14], given to relief symptoms of the disease.

The analysis of the urinary proteome, in particular that associated with the exosomes released into the lumen by tubular epithelial cells, provides useful and reliable information on the abundance of proteins expressed at the apical membrane of the tubular cells themselves, in physio-pathological or experimental conditions, in a safe, cost-effective, and non-invasive way [15,16,17,18]. Strikingly, we found that urinary excretion of AQP2 and NKCC2 was significantly increased, as early as the second week of treatment with mirabegron, and the increases persisted until the end of the observation period (12 weeks).

Although in these subjects the mirabegron-induced increase in membrane expression of AQP2 and NKCC2 did not result in further urine concentration compared to baseline, and this was to be expected as they were not defective in the urine concentration mechanism, these results may represent proof-of-principle that in XNDI subjects β3-AR stimulation with mirabegron, at the proper dose, could promote antidiuresis and correct, albeit partially, polyuria.

## 2. Results

### 2.1. Localization of β3-AR in Human Kidney Tubules

Since we previously reported a detailed analysis of the localization and possible physiological function of β3-AR in the mouse kidney tubule [1,19], here we checked whether β3-AR has the same localization in human kidney samples. As shown in Figure 1A, immunofluorescence analysis performed on semi-thin cryosections of a human kidney indicated a basolateral expression of β3-AR (in red) in the kidney CD tubules, stained at the apical plasma membrane with anti-AQP2 antibodies (in green). As for NKCC2 (in green), the antibody against NKCC2 [20] detected the protein at the apical membrane of the TAL epithelial cells showing basolateral expression of β3-AR (Figure 1B, in red). To confirm the same localization of β3-AR that we showed in mice, we confirmed the absence of β3-AR staining in AQP1-positive kidney tubule (Figure 1C).

This evidence represented the fundamental prerequisite to hypothesize that treatment with the human β3-AR agonist mirabegron, could promote the apical trafficking of AQP2 and NKCC2 in humans and, in turn, increase the urinary excretion of both proteins via the apical exosome pathway.

### 2.2. Study in OAB Patients

#### 2.2.1. Study Design and Patients Demographics

As reported in Figure 2, in the enrollment phase, subjects underwent screening, physical examination and signed informed consent. Consented patients entered the run-in phase during which they were asked to collect the 24 h urine at the end of the first and second week of the run-in (T-1 and T0). At T0, they started the pharmacological treatment with mirabegron (Betmiga^®^, Astellas Pharma, Tokyo, Japan, see methods for details), and were asked to collect and deliver to the Urology unit the 24 h urine collection at weeks 1,2,4,8 and 12. All urine samples collected either before or after mirabegron treatment, were stored at −80 °C and processed to quantify urine output, urine osmolality, and urine excretion of Na^+^, K^+^, Cl^−^, creatinine, uAQP2 and uNKCC2.

A total of 50 patients with symptoms of OAB for ≥3 months were enrolled, of whom 44 completed the two weeks run-in phase and were included in the study (Figure 3). All 44 patients received mirabegron (Betmiga^®^ 50 mg/day), of whom 25 (cluster 1) provided urine sampling up to week 4 (T4). Of these, 20 patients (cluster 2) completed sampling up to week 8 (T8) and 16 patients (cluster 3) completed the study, providing the 24 h urine samples up to week 12 (T12). The number of patients who withdrew from the study at each time point is reported in Figure 3. Demographics for each cluster of patients are reported and no statistically significant difference related to gender or mean age was found within each cluster or between clusters.

#### 2.2.2. Urine Output and Osmolality before and throughout Treatment with Betmiga^®^

We measured the volume and the urine osmolality of the 24 h urine collection for each patient. Data are reported in the histogram in Figure 4. The three columns in darker gray report the results obtained in cluster 1 (n = 25), the group of subjects who provider urine samples during the run-in phase and up to the 4th week of Betmiga^®^ Treatment. The column in middle gray indicates values of urine output and osmolality in cluster 2 (n = 20), the cluster of subjects who were compliant with the study by providing samples in the run-in phase and then up to the eighth week of treatment with mirabegron. In lighter gray, the column relative to subjects belonging to cluster 3 (n = 16) who completed the entire sampling envisaged in our study. Data are reported as mean ± SEM and expressed as percentage of the values measured in baseline (mean value of T-1 and T0, set as 100%).

No statistically significant differences were seen in the urine output before and after Betmiga^®^ treatment, except for a non-statistically significant tendency to increased urine output at T12. Likewise, urine osmolality remained constant compared to the baseline up to T8. At T12, however, a small (around 10%), significant decrease of urine osmolality was observed in cluster 3, whose possible explanation will be discussed in the next session of the paper.

#### 2.2.3. Urine Electrolytes Excretion before and throughout Treatment with Betmiga^®^

Urine excretion of the principal electrolytes involved in the countercurrent mechanism of urine concentration, namely Na^+^, K^+^ and Cl^−^, was also measured throughout the study. As shown in Figure 5, no changes in electrolyte excretion were observed to be related to the Betmiga^®^ treatment. Additionally, for electrolytes, values are expressed as percentages of the excretion at baseline in the figure and reported as absolute values in Table 1. The trend of urinary electrolyte excretion showed no changes to be correlated with Betmiga treatment, neither if expressed as mEq/L, nor if normalized for urine output (mEq/24 h), or for urinary creatinine (mEq/mg creatinine) (Table 1).

#### 2.2.4. Urine Excretion of AQP2 and NKCC2 before and throughout Treatment with Betmiga^®^

The excretion of both AQP2 and NKCC2, quantified by ELISA test, normalized to the 24 h urine volume, and expressed as a percentage of the baseline values for each patient, is reported in Figure 6. Raw data (not shown) of AQP2 excretion at baseline are comparable to those shown by healthy subjects in other studies. The excretion of both proteins became statistically higher than the baseline reference already at the second week (T2) of Betmiga^®^ treatment (+44% for AQP2 and + 136% for NKCC2 at T2). Interestingly, uAQP2 remained about 30% above baseline, up to the 12th week of Betmiga^®^ administration. The increase of uNKCC2, compared to the baseline value, was evident and statistically significant up to the 8th week of treatment (T8). At week 12 (T12), although the mean value was about 180% of baseline, due to the high intra-group variability and the smaller sample size of cluster 3 (n = 16), the values were not statistically different from baseline.

## 3. Discussion

In recent years, we thoroughly investigated the renal localization of the β3-AR and the effect of its stimulation on tubular reabsorption of water and electrolytes [1,10,19]. Beyond the possibility of hypothesizing a novel regulatory mechanism of renal function by the sympathetic nervous system, which could corroborate the classic hormonal regulation of renal channels and transporters (by means of AVP, aldosterone, ANGII and others), this evidence has significant clinical implications.

In fact, along the renal tubule the β3-AR shares the same cellular localization and the same cAMP-mediated intracellular signaling, with the AVPR2, the G protein-coupled receptor, whose physiological stimulation by the AVP allows the kidney to operate the antidiuresis. In antidiuresis, the concerted activation/membrane translocation of key plasma membrane proteins such as AQP2 and NKCC2 allows the osmotic reabsorption of 99% of the enormous volume of glomerular filtrate (≈180 L/day). The fundamental role of AVPR2 in regulating the body’s water and electrolyte balance is highlighted by the human phenotype resulting from inactivating mutations of the receptor, which cause a rare congenital disease, X-linked nephrogenic diabetes insipidus (XNDI) characterized by exacerbated polyuria (up to 20 L of urine produced in 24 h) and constant risk of severe dehydration [21,22,23,24,25,26,27].

The presence of the β3-AR receptor on the same tubular cells expressing AVPR2, together with the availability of synthetic β3-AR agonists such as mirabegron (Betmiga^®^), the first approved for use in humans for the treatment of an overactive bladder (OAB) [11,13,28,29,30], suggest that mirabegron treatment might be effective in inducing antidiuresis in XNDI patients. We already obtained the proof-of-principle that treatment with a synthetic β3-AR agonist was able to restore antidiuresis in the mouse model of human XNDI with conditional deletion of the AVPR2. The next step, addressed in the present work, was to investigate whether, also in humans, the β3-AR agonism can positively affect the trafficking of AQP2 and NKCC2 in tubular cells in vivo. We exploited the possibility of obtaining preliminary indications in subjects with OAB under treatment with mirabegron. Of note, the subjects recruited and analyzed in this observational study have apparently normal urine concentrating ability. For this reason, the expected result was not to observe a supraphysiological concentration of urine, rather, an effect on the membrane trafficking of key transporters and channels involved in diuresis.

We verified that also in humans, as we have shown in mice, the receptor is expressed on the basolateral membrane of tubular cells expressing AQP2 and NKCC2 (Figure 1). The design of our study was intended to observe whether the treatment with mirabegron was able to induce changes in the urinary excretion of AQP2 and NKCC2 in OAB patients. Urine output, urine osmolality and urinary excretion of the main electrolytes were also monitored throughout the study. To minimize the effect of interindividual variability in urinary excretion of AQP2 and NKCC2, for each patient the baseline excretion levels (run-in phase) were set as 100% and the changes measured during treatment with Betmiga^®^ expressed as percentage of baseline. Compared to baseline, no significant changes in urine output were observed in the 12 weeks following the initiation of drug therapy with Betmiga^®^. However, a trend, although not statistically significant, towards an increase in 24 h urine output was observed at T12 (Figure 4). Additionally, the urine osmolarity did not change, except at the 12th week of treatment, at which a modest (10%) but significant decrease was observed (Figure 4). Similarly, the urine excretion of Na^+^, K^+^ and Cl^−^ did not change after the start of treatment with Betmiga^®^. To the best of our knowledge, changes in urine output, urine osmolarity and urine electrolytes excretion have never been observed or related to treatment with Betmiga^®^, in the clinical trials for drug validation. In fact, in a subject with normal renal function, an increase in tubular reabsorption of water and salts could predispose to blood hypertension and would be considered a serious side effect for a drug, which would discourage its use. The small reduction of urine osmolality that we observed at T12, could be a due to the reduction in urgency/incontinence episodes, and improvements in QoL [31], that might have induced patients to increase their intake of water. It should also be emphasized that the data at T12 came from the smallest cluster of patients (n = 16), and this could confer statistical instability to the results. On a higher number of patients, the difference may not be statistically significant.

However, the most interesting result of this observational study concerns the urine excretion of AQP2 and NKCC2. We confirmed that, also in humans, the stimulation of the β3-AR receptor on the renal tubule increases the plasma membrane expression of AQP2 and NKCC2. In mice, this effect was directly verified by immunolocalization experiments on kidney sections of mice treated with the murine β3-AR agonist BRL37344 [1]. In humans, for evident ethical reasons, it was not possible to pursue this approach. We used an alternative, non-invasive approach, based on the semiquantitative analysis of proteins expressed on the apical membrane which, upon entering the exosomes pathway, are excreted in the urine in an amount proportional to their abundance on the plasma membrane. The molecular composition of urinary exosomes reflects their cellular origin and for this reason they are considered a promising source of biomarkers of physio-pathological states [18,32]. Both AQP2 and NKCC2 were found by proteomic profiling of exosomes in human urine [33]. For AQP2, since the demonstration of the feasibility of this approach [15,34], a plethora of publications have positively correlated the membrane expression of AQP2 and its excretion via urinary exosomes. Likewise, a correlation between renal tissue expression of NKCC2 and its expression on urinary exosomes has been demonstrated [35,36,37].

The fact that mirabegron rapidly increases and maintains elevated levels of uAQP2 and uNKCC2 for at least 12 weeks of treatment, reinforces the possibility that β3-AR agonist stimulation may have an effect on XNDI patients. In fact, in these patients, the main mechanism triggering antidiuresis is compromised, due to inactivating mutations of AVPR2. The stimulation of β3-AR, given the same location and the same signal transduction mechanism as AVPR2, could partially restore the antidiuresis in these subjects. Importantly, we previously reported that human β3-AR is resistant to agonist-induced desensitization in renal cells [19], supporting the idea that its chronic stimulation could be exploited as a treatment of XNDI.

In this study, an increase in the urinary excretion of uAQP2 and uNKCC2, not paralleled by a decrease in the urinary excretion of water and salts, could appear as an apparent discrepancy. However, it should be considered that, in these OAB subjects with apparently normal renal function, the sole increase of NKCC2 at the plasma membrane, without a concomitant hyperactivation of Na/K-ATPase on the basolateral side, should not be sufficient to increase the transepithelial reabsorption of NaCl. In the absence of an increase in the tonicity of the medullary interstitium, even an increase in AQP2 at the plasma membrane should not promote further osmotic water reabsorption in the CD. On the other hand, in an XNDI subject, characterized by low AQP2 and NKCC2 at the plasma membrane and dissipated interstitial tonicity, treatment with mirabegron would: (a) increase NKCC2 in the membrane, restore NaCl reabsorption in the TAL, thus increasing the tonicity of the renal medulla, (b) increase AQP2 membrane expression in the CD, thus favoring water reabsorption, (c) ultimately rescue antidiuresis.

In conclusion, the evidence from this observational study, albeit on a small cohort of OAB patients, clearly indicates that treatment with mirabegron (Betmiga^®^) sustainably increases the urinary excretion of AQP2 and NKCC2, likely as a consequence of their membrane abundance in CD and TAL. Given their key role in triggering antidiuresis, this effect suggests that the treatment with mirabegron, or other β3-AR agonists approved for use in humans, might be a therapeutic tool to cure XNDI caused by inactivating mutations of AVPR2. Considering the safety and tolerability profile of mirabegron [11], already approved for human use, we suggest that a clinical trial, aimed at evaluating the recovery of antidiuresis on XNDI patients, would have a good chance of approaching a cure for this disease.

## 4. Materials and Methods

### 4.1. Human Kidney Immunofluorescence

Human kidney samples were embedded in optimal cutting temperature (OCT) medium and ultrathin cryosections (5 μm) placed on Superfrost/Plus Microscope Slides (www.fishersci.com, accessed on 22 November 2022). Human kidney cryosections were fixed with methanol at −20 °C for 10 min, treated with SDS 1% for 10 min and subjected to immunofluorescence analysis. Nonspecific binding sites were blocked with 1% bovine serum albumin in PBS for 30 min. Sections were then incubated with the primary antibodies rabbit anti-BAR3 (SC-50436, Santa Cruz Biotechnology, Inc., www.scbt.com, accessed on 22 November 2022) with mouse anti-AQP2 antibody [38] (SC-515770, Santa Cruz Biotechnology, Inc., www.scbt.com, accessed on 22 November 2022) or mouse anti-NKCC2 [20] or mouse anti-AQP1 [39,40] (SC-25287, Santa Cruz Biotechnology, Inc., www.scbt.com, accessed on 22 November 2022). Sections were incubated with Alexa Fluor-conjugated secondary antibody. Confocal images were obtained with a confocal laser-scanning microscope (Leica TSC-SP5, Leica, www.leica-microsystems.com, accessed on 22 November 2022). 

### 4.2. Participants and Study Design

A total of 50 Patients with overactive bladder (OAB) were recruited at the Urology, Andrology and Kidney Transplantation Unit, Department of Emergency and Organ Transplantation, University of Bari. All the procedures to monitor the patients and to obtain the biological materials proposed in this protocol are routinely performed at this Unit.

The detailed written protocol was submitted and approved by the Local Independent Ethical Committee of the Azienda Ospedaliero-Universitaria “Consorzio Policlinico” (study #4850, protocol #81130). A written informed consent was requested of all patients after full and detailed explanation of the aims of the study.

The experimental protocol is safe since it will be performed following standard guidelines carried out in accordance with the Helsinki Declaration. An accurate history was recorded and physical exam of each patient in the screening period was performed. Patient demographics (sex, race, age) and baseline urinary values (number of micturitions per 24 h, urine 24 h volume, creatininuria, electrolytes, osmolality) were recorded at the start of the two weeks run-in period. A diary was constructed for each patient including information on drug dosage, side effects, blood tests, additional information (smoking, glycemia, weight, side effects, etc.).

The inclusion criteria for patients were: OAB symptoms for ≥3 months (urinary frequency and urgency with or without urge incontinence); mean micturition frequency ≥8 times per 24 h during the 3-day micturition diary period; at least 3 episodes of urgency (grade 3 or 4) with or without incontinence, during the 3-day micturition diary period.

The exclusion criteria were: breastfeeding; pregnancy; stress incontinence; indwelling catheter or intermittent self-catheterization; Neurogenic Lower Urinary Dysfunction (LUTD); diabetic neuropathy; treatment with statins, diuretics, thiazolidinediones, metformin, sildenafil; symptomatic urinary tract infection, chronic inflammation such as interstitial cystitis, bladder stones, previous pelvic radiation therapy or previous or current malignant disease of the pelvic organs; non-drug treatment including electro-stimulation therapy; severe hypertension; hypersensitivity to anticholinergics, YM178 (mirabegron), other beta-adrenoreceptor (BAR) agonists, or lactose or any of the other inactive ingredients; treatment with any investigational drug or device within 30 days; average total daily urine volume >3000 mL as recorded in the 3-day micturition diary period; serum creatinine > 150 µmol/L, aspartate aminotransferase (AST) and/or alanine aminotransferase (ALT) > 2× upper limit of normal range (ULN), or gamma glutamyl transferase (γ-GT) > 3x ULN; clinically significant abnormal electrocardiogram (ECG). Withdrawal criteria: development of one or more of the conditions reported in ‘exclusion criteria’ during the study; withdrawal of informed consent; poor compliance; problems with urine sampling.

To evaluate the effect of mirabegron (Betmiga®, Astellas Pharma, Assago, MI, Italy) 50 mg/day on urine output, urine osmolality, urine excretion of electrolytes AQP2 and NKCC2, 24 h urine samples were collected at weeks −1 (T-1) and 0 (T0) before the initiation of Betmiga treatment and at weeks 1, 2, 4, 8 and 12 (T1, T2, T4, T8, T12) after the start of the treatment.

### 4.3. Number of Subjects

Using a significance level of 5% and a power of 95% it was calculated that the number of subjects needed in this study was 16, when the minimal relevant difference in uAQP2 and uNKCC2 was 0.3 and SD was 0.3. Analysis was performed with the G*Power software 3.1.

### 4.4. Urine Collection and Analyses

All urine samples from patients enrolled in this study were collected at the Department of Precision and Regenerative Medicine—Urology, Andrology and Kidney Transplantation Unit of University of Bari. The 24 h urine samples were supplemented with Protease Inhibitor Cocktail Tablets (Roche Diagnostics GmbH, Mannheim, Germany), centrifuged at 3000 rpm for 10minutes at 4 °C to remove cellular debris, and stored at −80 °C to preserve urine exosome integrity. Urine creatinine was measured enzymatically (IL Test TM Creatinine, Instrumentation Laboratories, Werfen, Bedford, MA, USA). Urine concentrations of sodium, potassium, chloride and creatinine were measured using routine methods at the Urology, Andrology and Kidney Transplantation Unit of University of Bari (Italy). Urine osmolality was measured using a vapor pressure osmometer (Vapro^®^ 5600; ELITechGroup, www.elitechgroup.com, accessed on 22 November 2022).

### 4.5. ELISA Test for uAQP2 and uNKCC2 on Urine Sample

Urine excreted AQP2 (uAQP2) was measured by an ELISA protocol originally established by Umenishi et al. [41], with some modifications [42]. Quantification of NKCC2 in urine samples (uNKCC2) was performed using a commercial ELISA kit (Catalog # MBS453691; https://biocheminfo.org, accessed on 22 November 2022). Data of uAQP2 and uNKCC2 excretion were expressed as picomoles and normalized to the 24 h diuresis (pmol/24 h).

### 4.6. Statistical Analysis

Statistical analyses were performed using GraphPad Prism 9. Baseline values were obtained by taking the average of the measurements from the two baseline time points (T-1 and T0) for each variable analyzed. The data we represented as means ± standard error mean (SEM). Statistical significance was defined for *p* < 0.05 in all analyses. Statistical analysis was performed using a one-way ANOVA test with Dunnett’s multiple comparison test.

## Figures and Tables

**Figure 1 ijms-24-01136-f001:**
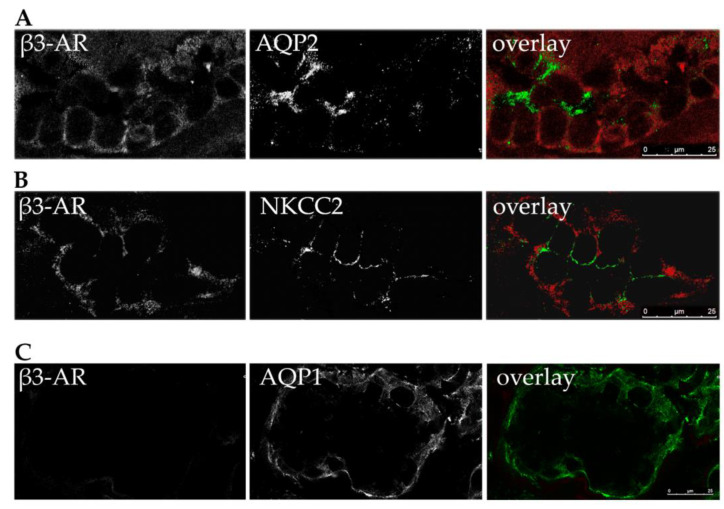
Localization of β3-AR in the collecting duct, the thick ascending Limb and the proximal tubule in human kidney. Human kidney sections were stained with anti β3-AR antibodies (in red) and co-stained with antibodies against (**A**) AQP2, (**B**) NKCC2 and (**C**) AQP1 (all in green). Overlay shows the expression of β3-AR at the basolateral membrane of cells expressing apical AQP2 (collecting duct), apical NKCC2 (thick ascending limb) but not in the AQP1-expressing tubules (proximal tubule/descending limb). Magnification bar = 25 µm.

**Figure 2 ijms-24-01136-f002:**
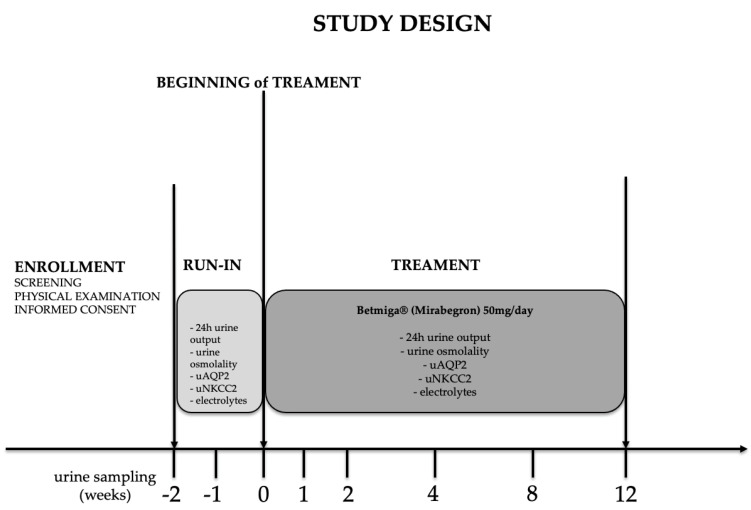
Design of the study. After initial screening, subjects were enrolled in the study. In the run-in phase, urine was evaluated at baseline at week −1 and 0 (T−1, T0). At week 0, patients initiated treatment with Betmiga^®^ 50 mg/day and urine were analyzed at weeks 1, 2, 4, 8 and 12 (T1–T12).

**Figure 3 ijms-24-01136-f003:**
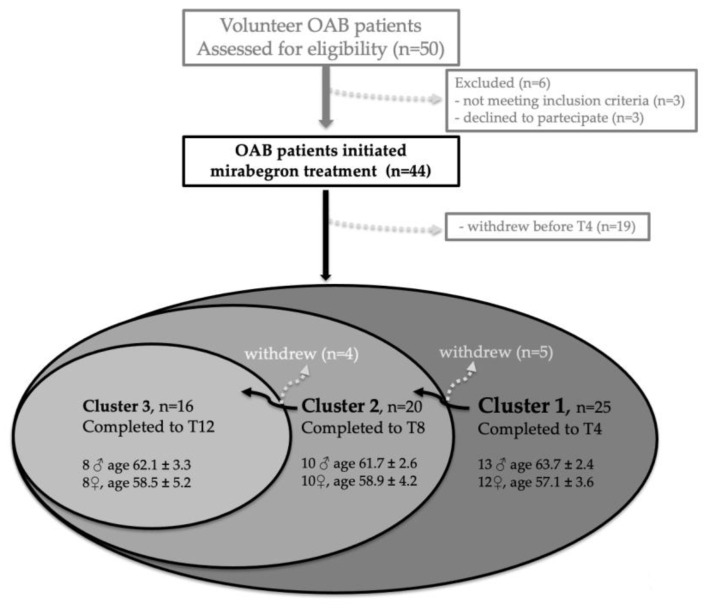
Compliance and demographics of patients participating in the study. After initial screening, 50 volunteer OAB patients were enrolled in the study. Six individuals were excluded or declined to participate. Forty-four patients initiated Betmiga^®^ treatment but only 25 (Cluster1) provided the 24 h urine collection at the end of the 1st, 2nd and 4th of treatment (T1, T2, T4). Of these, 20 (cluster 2) provided urine collections at the end of the 8th week of treatment (T8). Of these, 16 (cluster 3) provided urine collected at the end of the 12th week of treatment (T12). The clusters were homogeneous by gender and age and there was no statistically significant difference within each cluster or between them (one-way ANOVA with Dunnett’s multiple comparison test).

**Figure 4 ijms-24-01136-f004:**
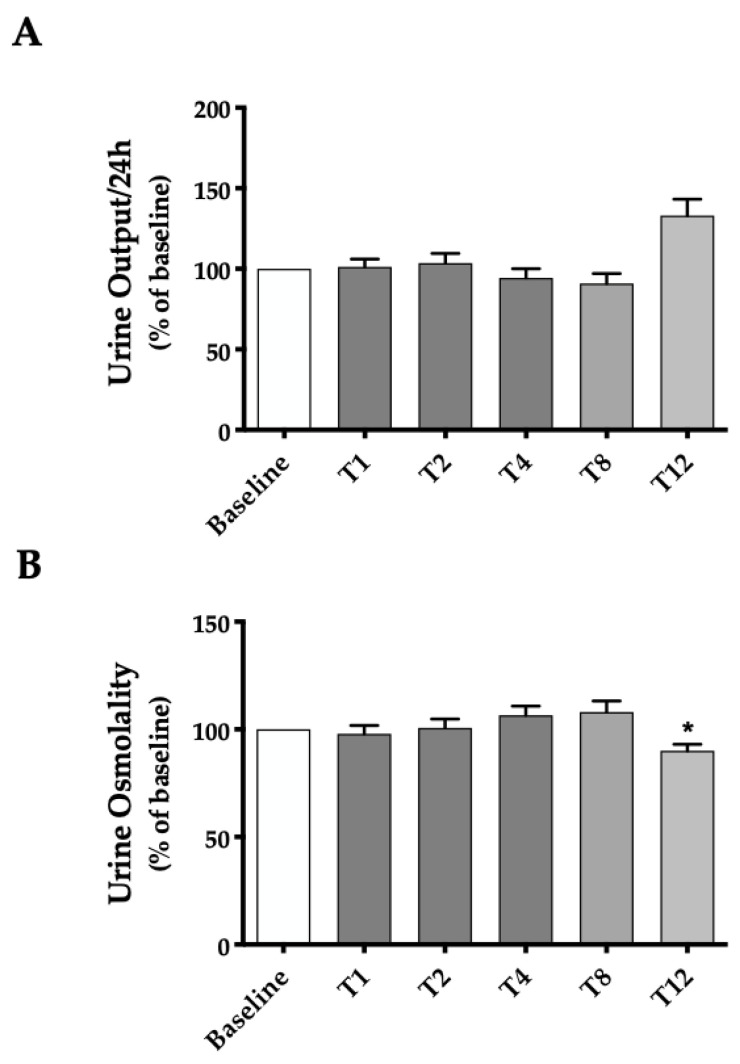
Urine output and osmolality before and during treatment with Betmiga^®^. Analysis was performed at baseline (T-1, T0) and at weeks 1, 2, 4, 8 and 12 (T1, T2, T4, T8, T12) after initiation of Betmiga^®^ treatment. (**A**) 24 h urine output was unchanged up to T12. (**B**) Urine osmolality was slightly decreased only at T12. Data are expressed as percent of values measured at baseline (set as 100%) for each patient. Shades of gray indicate different clusters of patients, as reported in Figure 3. The respective baseline was considered for each cluster. The values obtained were compared by one-way ANOVA with Dunnett’s multiple comparison test. (* *p* < 0.05).

**Figure 5 ijms-24-01136-f005:**
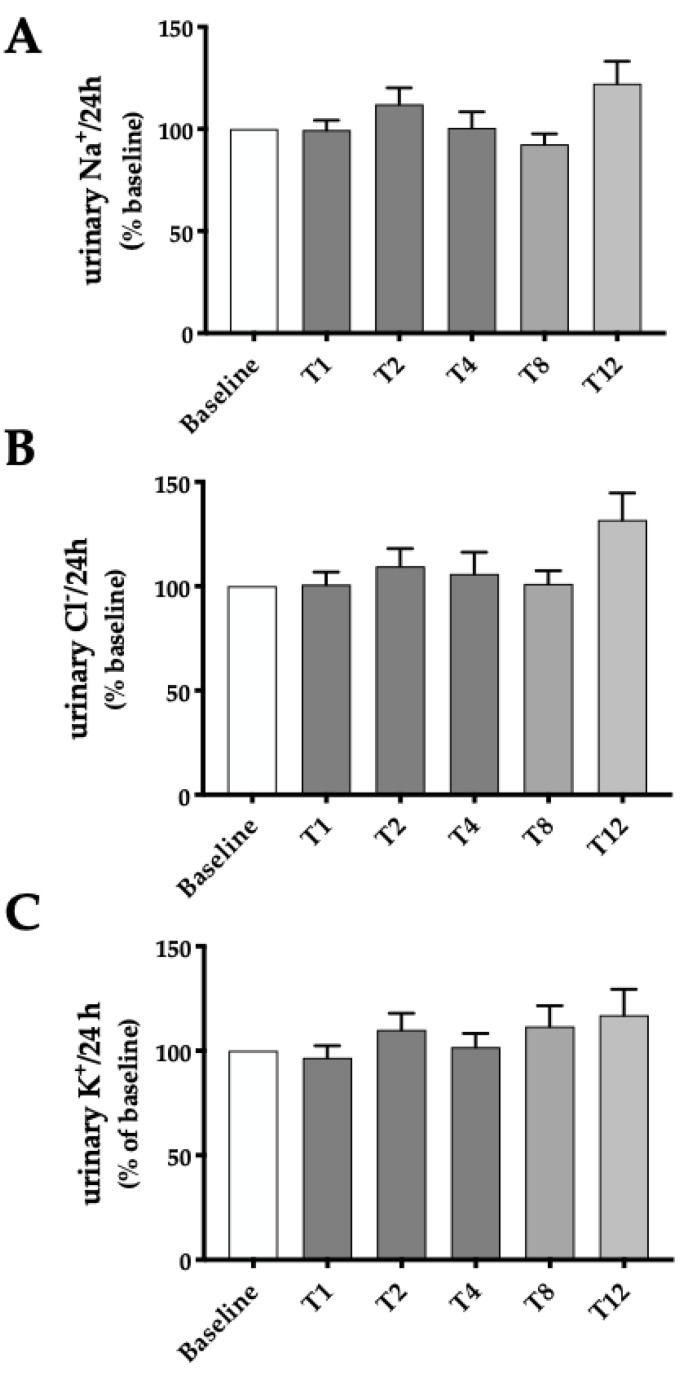
Urine electrolytes excretion before and throughout treatment with Betmiga^®^. Analysis was performed at baseline (T-1, T0) and at weeks 1, 2, 4, 8 and 12 (T1, T2, T4, T8, T12) after initiation of Betmiga^®^ treatment. (**A**–**C**) 24 h urinary excretion of Na^+^, K^+^ and Cl^−^ were unchanged up to T12. Data are expressed as percent of each concentration measured at baseline (set as 100%) for each patient. Shades of gray indicate different clusters of patients, as reported in Figure 3. The respective baseline was considered for each cluster. The values obtained were compared by one-way ANOVA with Dunnett’s multiple comparison test.

**Figure 6 ijms-24-01136-f006:**
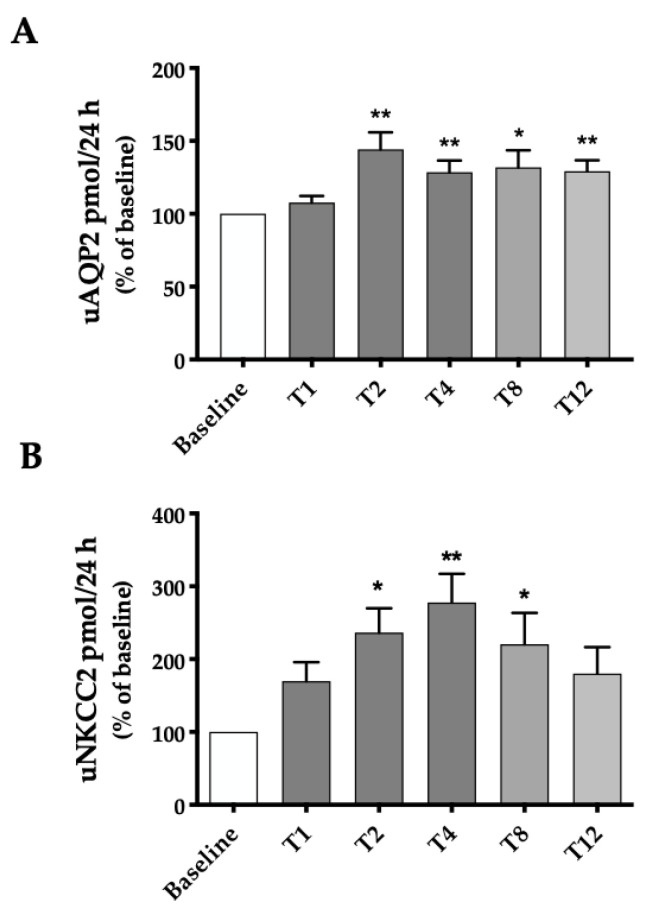
uAQP2 and uNKCC2 excretion before and throughout treatment with Betmiga^®^. Analysis was performed at baseline (T-1, T0) and at weeks 1, 2, 4, 8 and 12 (T1, T2, T4, T8, T12) after initiation of Betmiga^®^ treatment. (**A**) uAQP2 was significantly increased as early as T2 and remained about 30% above baseline, up to the 12th week of Betmiga^®^ administration. (**B**) uNKCC2 significantly increased at T2 and remained significantly higher than the baseline up to T8. Data are expressed as percent of values measured at baseline (set as 100%) for each patient. Shades of gray indicate different clusters of patients, as reported in Figure 3. The respective baseline was considered for each cluster. The values obtained were compared by one-way ANOVA with Dunnett’s multiple comparison test. (* *p* < 0.05; ** *p* < 0.01).

**Table 1 ijms-24-01136-t001:** Urinary parameters of patients enrolled in the study.

		Cluster 1, N = 25				Cluster 2, N = 20		Cluster 3, N = 16	
		Baseline	T1	T2	T4	Baseline	T8	Baseline	T12
**24 h Urine Output**	L	1.428 ± 0.161	1.314 ± 0.121	1.306 ± 0.109	1.220 ± 0.106	1.385 ± 0.162	1.130 ± 0.137	1.288 ± 0.183	1.419 ± 0.123
**Urine Osmolality**	mOsm/Kg	601.2 ± 40.7	589.2 ± 40.5	620.6 ± 47.8	644 ± 46.2	638.2 ± 42.2	699.4 ± 49.1	658.4 ± 45.9	596.4 ± 43.3 *
**Urine Na^+^**	mEq/L	115.3 ± 8.9	114.7 ± 10.5	118.8 ± 9.9	116.9 ± 10.5	122.4 ± 9.2	122 ± 12.5	122.1 ± 9.5	113.2 ± 8.8
	mEq/24 h	151.9 ± 17	139.7 ± 13	143.4 ± 10	132 ± 15	154.6 ± 18	137 ± 16	144.4 ± 19.3	157.1 ± 14
	mEq/mg Creatinine	0.103 ± 0.006	0.111 ± 0.007	0.107 ± 0.006	0.098 ± 0.007	0.10 ± 0.005	0.092 ± 0.007	0.102 ± 0.006	0.101 ± 0.006
**Urine K^+^**	mEq/L	44.3 ± 3.7	41.9 ± 4.2	45.2 ± 3.3	45.9 ± 3	45.01 ± 4.1	51.5 ± 4	47.5 ± 4.8	40.3 ± 3.3
	mEq/24 h	54.7 ± 5.7	48 ± 4.6	53.4 ± 4	50.1 ± 3.7	54.8 ± 6.1	54.5 ± 4.7	54.2 ± 7.7	54 ± 4.7
	mEq/mg Creatinine	0.038 ± 0.003	0.039 ± 0.003	0.04 ± 0.002	0.038 ± 0.002	0.04 ± 0.003	0.039 ± 0.002	0.04 ± 0.003	0.036 ± 0.002
**Urine Cl^−^**	mEq/L	116.5 ± 8	115.8 ± 9	123.6 ± 9	127.8 ± 11	120 ± 8.6	135.5 ± 13	119.7 ± 8.5	115.4 ± 9
	mEq/24 h	149.7 ± 17	137.3 ± 12	143 ± 11	137.9 ± 15	153.3 ± 18.2	145.9 ± 15	140 ± 19.1	156.4 ± 14
	mEq/mg Creatinine	0.101 ± 0.008	0.110 ± 0.007	0.109 ± 0.005	0.104 ± 0.006	0.1 ± 0.006	0.099 ± 0.006	0.1 ± 0.006	0.101 ± 0.006
**Urine Creatinine**	mg/ml	1.18 ± 0.09	1.15 ± 0.09	1.21 ± 0.09	1.22 ± 0.08	1.26 ± 0.10	1.29 ± 0.10	1.32 ± 0.12	1.18 ± 0.09
	mg/24 h	1450 ± 130	1240 ± 80	1350 ± 90	1350 ± 80	1470 ± 140	1400 ± 110	1490 ± 90	1570 ± 90

Data are present as mean ± SEM and compared by one-way ANOVA with Dunnett’s multiple comparison test. (* *p* < 0.05).

## Data Availability

Not applicable.

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
