# Peer review of "β3 Adrenergic Receptor Agonist Mirabegron Increases AQP2 and NKCC2 Urinary Excretion in OAB Patients: A Pleiotropic Effect of Interest for Patients with X-Linked Nephrogenic Diabetes Insipidus"

_ijms, 2023, doi:10.3390/ijms24021136_

Round 1
Reviewer 1 Report
In this work, Milano et al., evaluate the effect of the administration of the β3-AR agonist on the trafficking of renal transporters in a population of patients treated with mirabegron to relieve the symptoms of overactive bladder. In the past, the same authors showed the expression of the β3-AR receptor in the mouse kidney and demonstrated that the stimulation of this receptor triggers antidiuresis by promoting the apical expression of aquaporin 2 (AQP2) and Na-K-Cl cotransporter NKCC2. The fact that the antidiuretic effect of β3-AR receptor agonism evokes antidiuresis even in vasopressin receptor knockout mice, allowed the authors to postulate that stimulation of the β3-AR might be a therapeutic strategy to cure symptoms of the human disease nephrogenic diabetes insipidus, due to inactivating mutations of the V2R receptor for vasopressin.
The present work represents a step forward in the attempt to prove this thesis and aims to demonstrate that also in humans the stimulation of the β3-AR receptor affects the membrane trafficking of these two proteins.
They exploit the fact that mirabegron is approved for use in patients with OAB and build an experimental approach to analyze what happens in the renal tubule of these patients, by measuring the amount of AQP2 and NKCC2 in the urine of these patients.
The manuscript is clear, the results rigorously presented, and the conclusions adequately discussed. Increased urinary excretion of AQP2 and NKCC2 are evident and maintained throughout the monitoring period.
However, there are some aspects that could strengthen the conclusions of the work.
- Some drugs taken for the treatment of rather common disorders can have an effect on the traffic of the two transporters analyzed in the work. For example, metformin, a medication that is widely used to treat type 2 diabetes mellitus, showed an effect on AQP2, urea transporter UT-A1 and, consequently, an antidiuretic effect on the same model murine animal used by this research group (Klein et al., AJP, 20216) or in tolvaptan-treated rats (Efe et al, JCI insights, 2016). Similarly, sildenafil, the active ingredient of Viagra, by increasing intracellular cGMP levels, activates membrane translocation of AQP2 in vitro and in vivo in laboratory animals (Bouley et al., AJP renal, 2005) and in man (Assadi et al., Am J nephrol, 2015), inducing antidiuresis.
In the exclusion criteria of the study it is not specified if any of the patients are being treated with these drugs that could alter the trafficking of AQP2 and, consequently, the fluid balance of the patients. This aspect needs to be clarified.
- It has been demonstrated that in rodents the β3-AR receptor is expressed in several areas of the brain, including the hypothalamus (Claustre et al., Neuroscience, 2008) which is the seat of vasopressin neurons that secrete vasopressin into the circulation. Is there a possibility that mirabegron may have determined the effect on AQP2 and NKCC2 by stimulating the release of vasopressin in these patients? Was vasopressin measured in patients' plasma before and after initiation of Betmiga treatment to rule out this possibility?
- The number of patients included in the study is relatively small due to the low compliance of the volunteers enrolled in the study. Did the authors perfor a power calculation to determine the minimum sample size?
Minor
- spelling mistake line 131 50mg/die should be 50mg/da
Author Response
Comments and Suggestions for Authors
In this work, Milano et al., evaluate the effect of the administration of the β3-AR agonist on the trafficking of renal transporters in a population of patients treated with mirabegron to relieve the symptoms of overactive bladder. In the past, the same authors showed the expression of the β3-AR receptor in the mouse kidney and demonstrated that the stimulation of this receptor triggers antidiuresis by promoting the apical expression of aquaporin 2 (AQP2) and Na-K-Cl cotransporter NKCC2. The fact that the antidiuretic effect of β3-AR receptor agonism evokes antidiuresis even in vasopressin receptor knockout mice, allowed the authors to postulate that stimulation of the β3-AR might be a therapeutic strategy to cure symptoms of the human disease nephrogenic diabetes insipidus, due to inactivating mutations of the V2R receptor for vasopressin.
The present work represents a step forward in the attempt to prove this thesis and aims to demonstrate that also in humans the stimulation of the β3-AR receptor affects the membrane trafficking of these two proteins.
They exploit the fact that mirabegron is approved for use in patients with OAB and build an experimental approach to analyze what happens in the renal tubule of these patients, by measuring the amount of AQP2 and NKCC2 in the urine of these patients.
The manuscript is clear, the results rigorously presented, and the conclusions adequately discussed. Increased urinary excretion of AQP2 and NKCC2 are evident and maintained throughout the monitoring period.
However, there are some aspects that could strengthen the conclusions of the work.
R- We thank the Referee for the positive evaluation of our work and for the valid suggestions proposed to improve the quality of the manuscript.
- Some drugs taken for the treatment of rather common disorders can have an effect on the traffic of the two transporters analyzed in the work. For example, metformin, a medication that is widely used to treat type 2 diabetes mellitus, showed an effect on AQP2, urea transporter UT-A1 and, consequently, an antidiuretic effect on the same model murine animal used by this research group (Klein et al., AJP, 20216) or in tolvaptan-treated rats (Efe et al, JCI insights, 2016). Similarly, sildenafil, the active ingredient of Viagra, by increasing intracellular cGMP levels, activates membrane translocation of AQP2 in vitro and in vivo in laboratory animals (Bouley et al., AJP renal, 2005) and in man (Assadi et al., Am J nephrol, 2015), inducing antidiuresis.
In the exclusion criteria of the study it is not specified if any of the patients are being treated with these drugs that could alter the trafficking of AQP2 and, consequently, the fluid balance of the patients. This aspect needs to be clarified.
R- We would like to thank the Referee for this suggestion. In fact, some drugs commonly administered to a large number of patients can alter the trafficking and functionality of transporters and renal channels. In the study design phase we took into account all those drugs whose effect on AQP2 and NKCC2 has been demonstrated in the literature. In the revised version of the manuscript (lines 349-50) we specified that therapies with metformin, statins, and sidenafil were present in the exclusion criteria for patients enrollment.
- It has been demonstrated that in rodents the β3-AR receptor is expressed in several areas of the brain, including the hypothalamus (Claustre et al., Neuroscience, 2008) which is the seat of vasopressin neurons that secrete vasopressin into the circulation. Is there a possibility that mirabegron may have determined the effect on AQP2 and NKCC2 by stimulating the release of vasopressin in these patients? Was vasopressin measured in patients' plasma before and after initiation of Betmiga treatment to rule out this possibility?
R- The doubt raised by the Referee can be fully shared and we have already tried to answer this question in a previous study in which we demonstrated the expression of the β3-AR receptor in the mouse kidney and its possible physiological role in the regulation of water balance (Procino et al Kidney Int., 2016). The fact that, in mice, stimulation of β3-AR with the synthetic agonist BRL37344 promotes AQP2 and NKCC2 trafficking and antidiuresis, even in vasopressin receptor (AVPR2) knockout mice, strongly suggests that the effect of β3-AR is not mediated by the release of vasopressin, to which these animals are not sensitive. Furthermore, in viable mouse kidney slices, pharmacological agonism of β3-AR promotes trafficking and activation of AQP2 and NKCC2, providing further evidence that stimulation of β3-AR in the kidney triggers its effect independently of stimulation of β3-AR expressed in the CNS. This aspect is now mentioned in the discussion of the revised manuscript.
In this study it was not possible to measure vasopressin in the patients' plasma as the study did not include blood sampling. In order to promote the highest possible compliance of volunteers, invasive practices such as blood sampling were avoided.
- The number of patients included in the study is relatively small due to the low compliance of the volunteers enrolled in the study. Did the authors perfor a power calculation to determine the minimum sample size?
R- We did perform power calculation, the results are now reported in the revised manuscript.
Minor
- spelling mistake line 131 50mg/die should be 50mg/da
R- Done
Reviewer 2 Report
The manuscript by Milano et al. entitled “β3 adrenergic agonist Mirabegron increases AQP2 and NKCC2 urinary excretion in OAB patients: a pleiotropic effect of interest for patients with X-linked Nephrogenic Diabetes Insipidus (reference ijms-2087268)” reports on the treatment with β3-AR agonist mirabegron on AQP2 and NKKC2 trafficking in a cohort of patients with overactive bladder syndrome. Mirabregon significantly increased urinary AQP2 an NKCC2 (this biomarker only for 8 weeks) for 12weeks of treatment in OAB patients. This may be of clinical relevance for the treatment of XNDI patients. Treatment with mirabregon may bypass the inactivation of AVPR2 in XNDI patients, trigger antidiuresis and correct the dramatic polyuria which is the main hallmark of this disease. In the present manuscript the group replicates the immunofluorescence data obtained from mouse origin in human renal tissue. In OAB patients the group studied the effect of mirabregon (betmiga treatment) on urine output, urine osmolality and urinary Na+, K+ plus Cl- excretion. Expected trends were observed in this part of the study. Betmiga treatment significantly increased the urinary excretion of AQP2 and NKCC2 (only for 8 weeks) during the 12 week protocol in OAB patients. This reinforces that β3-AR agonist stimulation may have an effect in XNDI patients. This provides potential as for this group of patients. Next issues need tob e considered for improvement of this manuscript:
1.)The current manuscript may provide a nice example for drug repurposing in rare human disease. Mirabregon has already be approved for human use. Are there alternatives for betmiga in this group of drugs for treatment?
2.)Authors suggest a potential future clinical trial for XNDI patients with betmiga: Can you elaborate on how to execute this clinical trial (in your discussion)?
3.)Authors should provide experimental details for how the preparation of exosomes has been performed for the urinary samples of the OAB patients.
4.)Is the baseline control level (or control value) for urinary AQP2 an NKKC2 in OAB patients from this study similar to normal human controls?
5.)Can you provide the raw data for the ELISA results depicted in figure 6? Can an extra table containing these values be included?
Author Response
Comments and Suggestions for Authors
The manuscript by Milano et al. entitled “β3 adrenergic agonist Mirabegron increases AQP2 and NKCC2 urinary excretion in OAB patients: a pleiotropic effect of interest for patients with X-linked Nephrogenic Diabetes Insipidus (reference ijms-2087268)” reports on the treatment with β3-AR agonist mirabegron on AQP2 and NKKC2 trafficking in a cohort of patients with overactive bladder syndrome. Mirabregon significantly increased urinary AQP2 an NKCC2 (this biomarker only for 8 weeks) for 12weeks of treatment in OAB patients. This may be of clinical relevance for the treatment of XNDI patients. Treatment with mirabregon may bypass the inactivation of AVPR2 in XNDI patients, trigger antidiuresis and correct the dramatic polyuria which is the main hallmark of this disease. In the present manuscript the group replicates the immunofluorescence data obtained from mouse origin in human renal tissue. In OAB patients the group studied the effect of mirabregon (betmiga treatment) on urine output, urine osmolality and urinary Na+, K+ plus Cl-excretion. Expected trends were observed in this part of the study. Betmiga treatment significantly increased the urinary excretion of AQP2 and NKCC2 (only for 8 weeks) during the 12 week protocol in OAB patients. This reinforces that β3-AR agonist stimulation may have an effect in XNDI patients. This provides potential as for this group of patients. Next issues need tob e considered for improvement of this manuscript:
1.)The current manuscript may provide a nice example for drug repurposing in rare human disease. Mirabregon has already be approved for human use. Are there alternatives for betmiga in this group of drugs for treatment?
R- We are pleased that the Referee appreciated the importance of this proof of principle in the prospect of a cure for XNDI. We thank him/her for the suggestions to improve the quality of the manuscript. We have done our best to revise the manuscript in accordance with these suggestions.
Therapeutic options to correct OAB-related symptoms are the use of antimuscarinics or β3-AR agonists (see doi: 10.1590/S1677-5538.IBJU.2021.99.12 for review). From the perspective of exploiting the ability of β3-AR to induce antidiuresis in NDI, we are interested in the second option. Regarding β3-AR agonists, current guidelines of all scientific organizations strongly recommend mirabegron for the treatment of idiopathic OAB. More recently, after successful Phase III trials, a second β3-adrenergic receptor agonist, vibegron, was introduced in the Japanese and North American markets for OAB treatment. Other molecules, such as Amibegron and Ritobegrion have been discontinued due to adverse events encountered during Phase II and III trials. Solabegron has passed Phase II clinical trials but, to the best of our knowledge, is not yet commercially available for the treatment of OAB (see doi:10.3390/cells8040357 for review).
Our observational study was done with the most widely used β3-AR agonist to date. If the strategy we hypothesize is successful in NDI patients, all present and future β3-AR agonists available on the market could be tested for their efficacy in correcting NDI-associated polyuria.
2.)Authors suggest a potential future clinical trial for XNDI patients with betmiga: Can you elaborate on how to execute this clinical trial (in your discussion)?
R- We thank the Referee for the suggestion. The primary objective of a clinical trial must be to verify the ability of mirabegron or other β3-AR agonists to restore antidiuresis in NDI patients. Details on how to conduct this trial depend on the number of patients recruited. Since the XNDI pathology is a rare disease, the number of volunteers recruited will be important to define the type of trial, whether single-center, multicenter, with or without a placebo. Not having the possibility to hypothesize a number of volunteer NDI patients, it seems premature to discuss these details in the paper. Rather, we share this information with the scientific community and with those XNDI reference centers that may be able to gather enough patients to initiate experimentation.
3.)Authors should provide experimental details for how the preparation of exosomes has been performed for the urinary samples of the OAB patients.
R- The presence of urinary exosomes containing AQP2 and NKCC2 has been demonstrated by many studies from our and other research groups (DOI: 10.1073/pnas.0403453101; DOI: 10.1046/j.1523-1755.2002.00686.x; DOI: 10.1186/1471-2369-15-101). The abundance of these exosomes in the urine implies that it is not necessary to proceed with an isolation step, therefore the urine can be analyzed as it is using a 96-well plate ELISA assay, which has the advantage of having high sensitivity and high throughput. The isolation of exosomes is necessary if one wishes to proceed with the analysis of their proteome by immunoblotting or to study the ultrastructure of these exosomes by TEM. Some years ago, in a work that analyzed the urinary excretion of AQP2 in relation to the staging of diabetic nephropathy, we demonstrated that the amount of AQP2 analyzed in unprocessed urine by ELISA corresponds to the amount of AQP2 present in exosomes isolated from the same urine by differential centrifugation (https://doi.org/10.1155/2017/4360357). In the present work, therefore, the measurement of AQP2 and NKCC2 in urinary exosomes was performed on total urine, without proceeding with the isolation of urinary exosomes.
4.)Is the baseline control level (or control value) for urinary AQP2 and NKKC2 in OAB patients from this study similar to normal human controls?
R- Comparing the urinary excretion of AQP2 of OAB subjects before betmiga treatment (T-1 and t0) with those of other studies in which we had healthy control patients (around 1000 fmoles/mg creatinine), we can state that OAB subjects have urinary AQP2 excretion values comparable to those of healthy subjects. NKCC2 excretion in AOB patients is consistent with other observations (DOI: 10.1186/1471-2369-15-101). It should be underlined, however, that values of the urinary excretion of AQP2 show a high inter-individual variability, influenced by age, gender, hydration status, and lifestyle. This observation is consistent in our studies, and is the main reason why we prefer to show changes in AQP2 excretion as percentages and not as absolute values.
5.)Can you provide the raw data for the ELISA results depicted in figure 6? Can an extra table containing these values be included?
R- As previously mentioned, the interindividual variability of baseline AQP2 and NKCC2 excretion is not negligible. For this reason, we analyzed the phenomenon using percentage calculated considering the baseline (mean of values measured at t-1 and t0) as 100 % for each patient and believe that showing the raw data could lead to confusion for readers. Although the OAB cohort showed variable levels of urinary excretion of AQP2 and NKCC2 before and during treatment with mirabegron, we measured comparable percentage increases between individuals.
Round 2
Reviewer 2 Report
For me the revised version of the manuscript is acceptable for publication in a future issue of IJMS.